# Disease clusters subsequent to anxiety and stress-related disorders and their genetic determinants

Xin Han [1,2,3,12], Qing Shen [4,5,6,12], Can Hou [2,3], Huazhen Yang[2,3], Wenwen Chen[2,3], Yu Zeng[2,3], Yuanyuan Qu[2,3], Chen Suo[7,8], Weimin Ye [9], Fang Fang [6], Unnur A. Valdimarsdóttir [6,10,11,13] & Huan Song [2,3,10,13] ✉

Anxiety/stress-related disorders have been associated with multiple diseases, whereas a comprehensive assessment of the structure and interplay of subsequent associated diseases and their genetic underpinnings is lacking. Here, we first identify 136, out of 454 tested, medical conditions associated with incident anxiety/stress-related disorders attended in specialized care using a population-based cohort from the nationwide Swedish Patient Register, comprising 70,026 patients with anxiety/stress-related disorders and 1:10 birth year- and sex-matched unaffected individuals. By combining findings from the comorbidity network and disease trajectory analyses, we identify five robust disease clusters to be associated with a prior diagnosis of anxiety/stress-related disorders, featured by predominance of psychiatric disorders, eye diseases, ear diseases, cardiovascular diseases, and skin and genitourinary diseases. These five clusters and their featured diseases are largely validated in the UK Biobank. GWAS analyses based on the UK Biobank identify 3, 33, 40, 4, and 16 significantly independent single nucleotide polymorphisms for the link to the five disease clusters, respectively, which are mapped to several distinct risk genes and biological pathways. These findings motivate further mechanistic explorations and aid early risk assessment for cluster-based disease prevention among patients with newly diagnosed anxiety/stress-related disorders in specialized care.

Anxiety and stress-related disorders are among the most common mental disorders, with a regional variation in prevalence globally (i.e., from 5.3% to 10.4%)[1] and a pooled lifetime prevalence of ~12.9%[2]. With shared clinical symptoms and neurobiological features, anxiety and stress-related disorders are considered highly correlated——they are historically in the same diagnosis category[3], with similarities in genetic architecture demonstrated in familial coaggregation[4,5] and genome-wide association analysis (GWAS) studies[6].

[1]Mental Health Center, West China Hospital, Sichuan University, Chengdu, China. [2]West China Biomedical Big Data Center, West China Hospital, Sichuan University, Chengdu, China. [3]Med-X Center for Informatics, Sichuan University, Chengdu, China. [4]Clinical Research Center for Mental Disorders, Shanghai Pudong New Area Mental Health Center, Tongji University School of Medicine, Shanghai, China. [5]Institute for Advanced Study, Tongji University, Shanghai, China. [6]Institute of Environmental Medicine, Karolinska Institutet, Stockholm, Sweden. [7]Department of Epidemiology & Ministry of Education Key Laboratory of Public Health Safety, School of Public Health, Fudan University, Shanghai, China. [8]Taizhou Institute of Health Sciences, Fudan University, Taizhou, China. [9]Department of Medical Epidemiology and Biostatistics, Karolinska Institutet, Stockholm, Sweden. [10]Center of Public Health Sciences, Faculty of Medicine, University of Iceland, Reykjavík, Iceland. [11]Department of Epidemiology, Harvard T H Chan School of Public Health, Boston, MA, USA. [12]These authors contributed equally: Xin Han, Qing Shen. [13]These authors jointly supervised this work: Unnur A Valdimarsdóttir, Huan Song. ✉e-mail: songhuan@wchscu.cn

Individuals with anxiety and stress-related disorders usually follow an intermittent recurring symptom episode throughout life and a as result, experience impaired mental and physical functioning, increased rates of disability, and higher-than-expected mortality[7,8]. The causes of the excess disability and mortality include a considerable range of medical conditions, such as other psychiatric disorders (e.g., depression)[9], metabolic diseases[10,11], cardiovascular disease[12–14], auto-immune disease[15,16], and infections[17]. However, previous studies have mostly focused on specific groups of diseases and, to the best of our knowledge, no comprehensive assessment of disease clusters arising subsequent to anxiety and stress-related disorders has been conducted.

Although still early on, the scant literature suggests a role of stress-related genetic loci in the development of cardiovascular disease[18] and mortality[19] after exposure to terrorist attacks or social adversity. Likewise, a recent GWAS study revealed that both anxiety and stress-related disorders are genetically correlated with multiple obesity-related phenotypes[6], promoting studies on the genetic basis of adverse health consequences after anxiety and stress-related disorders. However, as a wide range of medical conditions have been associated with a prior diagnosis of anxiety and stress-related disorders, exploration of patients' genetic susceptibilities to a single disease has limited clinical implications. The recent advances in human disease network methodology, e.g., disease trajectory[20] and comorbidity network[21], provide new means to comprehensively summarize the possible sets of diseases (i.e., disease clusters with close temporal or non-temporal relationships) following a predetermined phenotype[22,23]. Furthermore, with the notion that disease located in the same cluster should have shared or linked biological mechanisms, the identification of cluster-specific, instead of diseases-specific, genetic variants, has the potential to realize the prevention of a general further health decline among patients with anxiety or stress-related disorders.

Therefore, taking advantage of the nationwide population and health registers in Sweden as well as the community-based health records and genetic information available in the UK Biobank, we aimed to identify major clusters of subsequent medical conditions after a diagnosis of anxiety and stress-related disorders. We further aimed to elucidate the underlying genetic determinants associated with those identified disease clusters.

## Results

### Baseline characteristics

Based on the specialized diagnoses from the nationwide Swedish Patient Register, we first included a Swedish cohort comprising 70,026 patients at first diagnosis of anxiety or stress-related disorders from 2001 to 2016 and ten randomly selected birth year- and sex-matched unaffected individuals per patient using incidence density sampling ($N = 700,260$), without history of other psychiatric disorders and severe somatic diseases, as the exploratory dataset (Fig. 1, Supplementary Fig. 1, Supplementary Data 1). To validate the identified disease clusters in the Swedish cohort, we constructed a UK cohort (i.e., the validation dataset) based on the UK Biobank, including 23,365 patients diagnosed with anxiety or stress-related disorders between 1997 and 2019 from inpatient and primary care, and 233,596 unaffected participants individually matched by sex and year of birth (Supplementary Fig. 2). The Swedish and the UK cohorts were both overrepresented by females (62.7% and 67.3% respectively), with a similar median age at diagnosis for anxiety and stress-related disorders (Table 1). The median follow-up time was 7.1 and 13.3 years in the Swedish cohort and UK cohort, respectively. Patients in both cohorts had lower educational and income levels than their matched unaffected individuals.

### Identification of associated disease clusters

In the Swedish cohort, 183 medical conditions (among 454 tested) had a prevalence of ≥0.5% and 136 were positively associated with a prior diagnosis of anxiety or stress-related disorders (Supplementary Data 2). The top HRs were noted for other psychiatric disorders, including personality and behavior disorder (HR [95% confidence intervals]: 16.7 [15.0–18.5]), sedatives or hypnotics abuse (16.0 [14.1–18.2]), and other mood disorder (13.4 [12.1–14.9]). Somatic medical conditions with the highest HRs were other headache syndromes (2.8 [2.6–3.0]), irritable bowel syndrome (2.6 [2.3–2.8]), and other functional intestinal disorders (2.3 [2.2–2.4]). In addition, one negative association was noted between anxiety/stress-related disorders and varicose veins of lower extremities (0.93 [0.89–0.98]).

Subsequently, we identified 433 and 97 disease pairs to construct a comorbidity network and disease trajectory after anxiety or stress-related disorders, respectively (details of the disease pair identification in Supplementary Fig. 3 and results in Supplementary Data 3-4). The analysis of the comorbidity network identified seven modules, characterized by their predominant components related to psychiatric disorders, eye diseases, ear diseases, cardiovascular disease, genitourinary diseases, musculoskeletal diseases, and cerebrovascular diseases (Fig. 2A). Figure 2B shows an overview of the disease trajectories after anxiety or stress-related disorders. The medical conditions that were listed immediately after anxiety or stress-related disorders in chronological order (i.e., D1) included disorders of sense organs, genitourinary diseases, cardiovascular disease, and psychiatric disorders.

We determined stable disease clusters by merging the findings from the comorbidity network and disease trajectory analyses together (Fig. 3), which consequently led to five clusters including 31 medical conditions (Fig. 4A, B). Cluster 1 denoted a link from anxiety or stress-related disorders to depression and alcohol abuse, and further to obesity and several other psychiatric disorders. Cluster 2 and Cluster 3 denoted a link to eye and ear diseases. Cluster 4 included mainly cardiovascular disease, with a direct link to hypertensive disorders, ischemic heart diseases, and angina pectoris. Cluster 5 was predominated by genitourinary diseases as the first affected medical conditions, and further linked to skin diseases.

Among the 31 medical conditions identified from the Swedish cohort, 19 medical conditions were validated in the UK cohort (i.e., statistically significantly associated with a prior diagnosis of anxiety or stress-related disorders, Supplementary Data 5). When validating the disease clusters, we identified 170 possible disease pairs and selected 55 disease pairs to be included in the comorbidity network analysis (Supplementary Data 6). All five disease clusters were replicated in the UK cohort, although the cluster predominated by psychiatric disorders and ear diseases from the Swedish cohort was found to be merged as one cluster in the UK cohort (Supplementary Table 1).

### Genetic determinants of associated disease clusters

To identify the potential genetic determinants for each disease cluster among patients with anxiety or stress-related disorders, we first calculated five cluster-specific quantitative scores as an index of individual's susceptibility to each disease cluster, and then performed GWAS analyses for the five susceptibility scores separately, among individuals from the UK cohort with eligible genotyping data ($n = 27,781$, Supplementary Fig. 2), using mixed linear model (MLM)-based models. GWAS analyses identified three, 33, 40, 4, and 16 independent single nucleotide polymorphisms (SNPs) for clusters featured by psychiatric disorders, eye diseases, ear diseases, cardiovascular diseases, and skin and genitourinary diseases, respectively (Table 2). The full lists of the SNPs, their mapped genes, and enriched biological pathways are presented in Supplementary Data 7–9. According to the genomic inflation analysis results of Linkage Disequilibrium (LD) score regression, we found little indication for confounding effects in the GWAS of the five

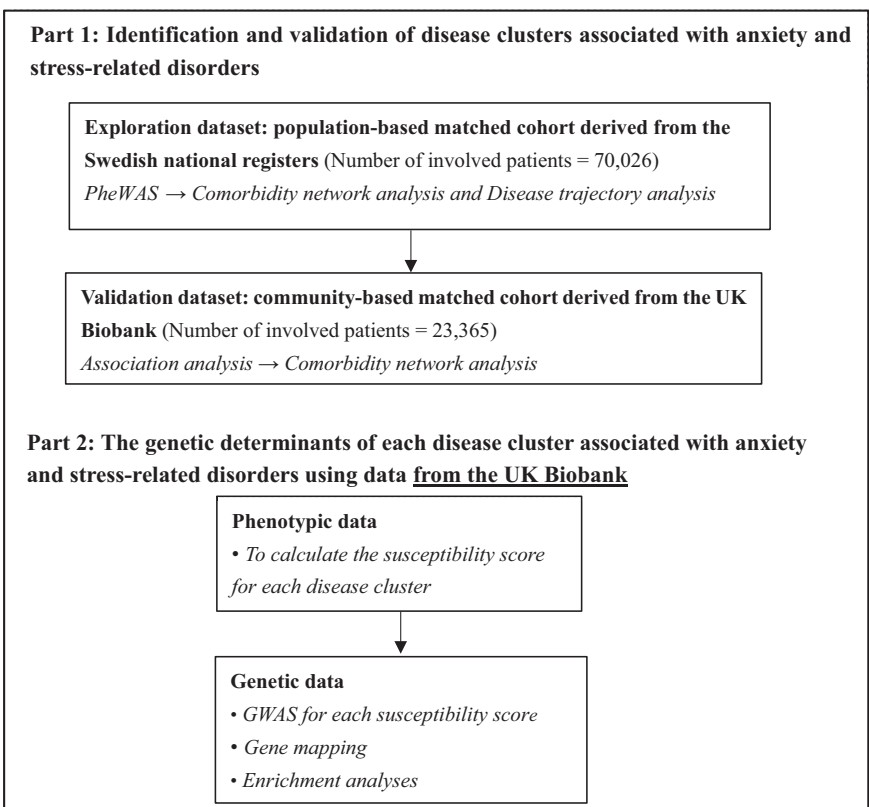

**Fig. 1 | Study design.** PheWAS phenome-wide association study, GWAS Genome-wide association study.

## Table 1 | Baseline characteristics of study participants from Swedish exploration cohort and UK validation cohort

| | Exploration dataset derived from Swedish national registers[a] | | Validation dataset derived from the UK Biobank | |
|---|---|---|---|---|
| | Individuals with anxiety and stress-related disorders (N = 70,026) | Matched individuals[b] (N = 700,260) | Individuals with anxiety and stress-related disorders (N = 23,365) | Matched individuals[b] (N = 233,596) |
| Age at diagnosis in years, median (IQR) | 51 (45–60) | 51 (45–60) | 52 (46–59) | 52 (46–59) |
| Follow-up time in years, median (IQR) | 7.1 (3.3–10.7) | 7.3 (3.5–10.8) | 13.5 (9.4–17.3) | 13.3 (9.4–17.3) |
| Sex (%) | | | | |
| Male | 26,093 (37.3) | 260,930 (37.3) | 7631 (32.7) | 76,285 (32.7) |
| Female | 43,933 (62.7) | 439,330 (62.7) | 15,734 (67.3) | 157,311 (67.3) |
| Highest Education (%) | | | | |
| <9 y | 13,816 (19.7) | 130,074 (18.6) | 12,250 (52.4) | 116,528 (49.9) |
| 9–12 y | 33,516 (47.9) | 329,046 (47.0) | | |
| >12 y | 22,694 (32.4) | 241,140 (34.4) | 6792 (29.1) | 79,987 (34.2) |
| Missing | | | 4323 (18.5) | 37,081 (15.9) |
| Income (%) | | | | |
| Lowest 20% | 1665 (2.3) | 14,917 (2.1) | 5046 (21.6) | 46,283 (19.8) |
| Middle | 50,210 (71.7) | 445,510 (63.6) | 13,968 (59.8) | 139,967 (59.9) |
| Highest 20% | 18,126 (25.9) | 238,475 (34.1) | 4314 (18.5) | 47,066 (20.2) |
| Missing | 25 (0.1) | 1358 (0.2) | 37 (0.1) | 280 (0.1) |

This table shows the baseline characteristics of the patients with anxiety and stress-related disorders and their matched unexposed individuals.

*IRQ* interquartile range.

[a]The Swedish cohort was presented only in third age group. The complete Swedish cohort was presented in Supplementary Table 13.

[b]At most 10 participants who were alive, retained in the cohort and free of anxiety and stress-related disorders at the corresponding index date were individually matched to each individual with anxiety and stress-related disorders based on sex, and year of birth.

A Comorbidity network after anxiety and stress-related disorders

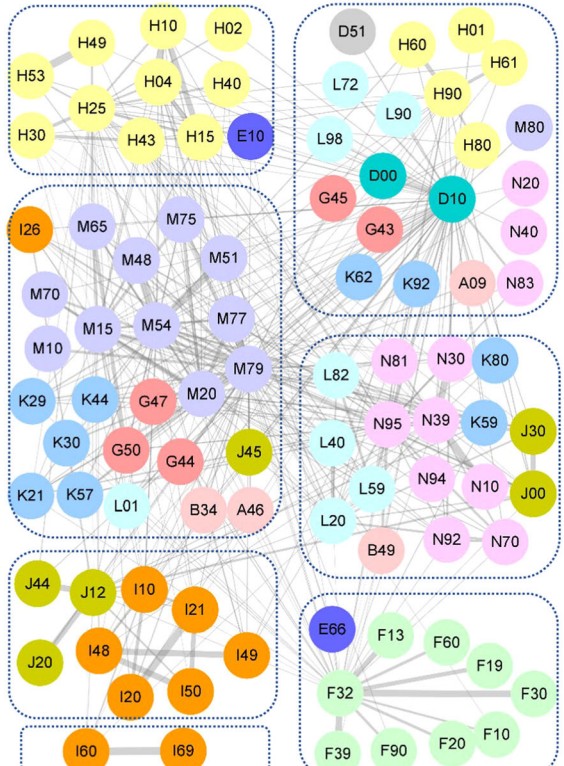

B Disease trajectory after anxiety and stress-related disorders

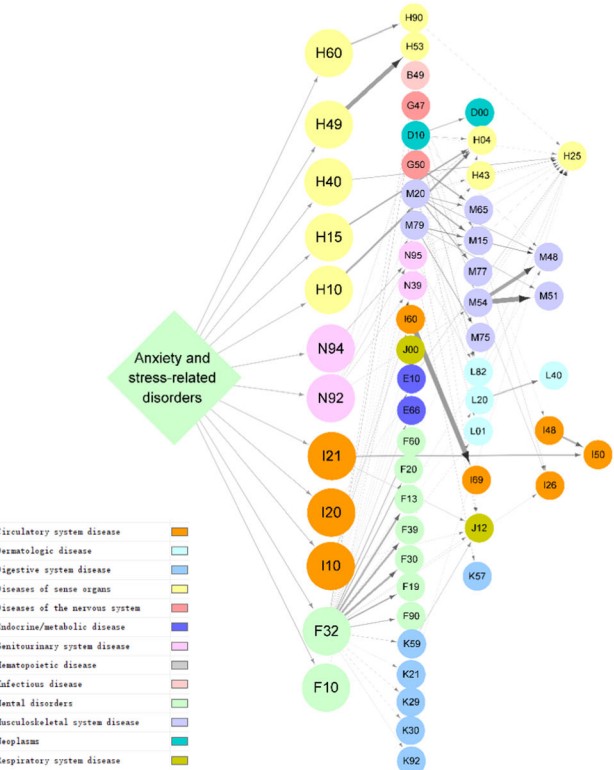

**Fig. 2 | Comorbidity network and disease trajectory after diagnosis of anxiety and stress-related disorders.** Each node represents a medical condition, and the combined ICD-10 codes are shown within the circle, while the color of the node indicates the category of the corresponding medical condition. The color of the link represents the strength of comorbidity association, measured by odd ratio. Definition of combined ICD-10 codes can be found in Supplementary Data 1.

**A** Comorbidity network after anxiety and stress-related disorders. The network was partitioned into seven main modules using Louvain algorithm, and nodes belonging to the same group were gathered together using blue dashes. **B** Disease trajectory after anxiety and stress-related disorders. The first affected diseases (D1) by anxiety and stress-related disorders included disorders of sense organs, genitourinary diseases, cardiovascular diseases, and psychiatric disorders.

disease clusters (Supplementary Table 2). We found 20 (e.g., *AP000304.12, ATP5O, MRPS6*), 362 (e.g., *ZFAND1, CHMP4C, SNX16*), 440 (e.g., *C10orf88, PSTK, TEX36*), 30 (e.g., *EFNA5, VAV3, SLC25A24*) and 229 (e.g., *BCHE, ZBBX, OR6S1*) mapped genes for the five associated disease clusters, which were then associated with several enrichment biologic pathways topped by GO:0098660 inorganic ion transmembrane transport, WP5224 2q37 copy number variation syndrome, GO:0048545 response to steroid hormone, M5884 NABA CORE MATRISOME, and GO:0097484 dendrite extension, respectively. Based on information from FUMA and GeneCards, we found that several cluster-specific genes have been previously associated with individual psychiatric or somatic traits in the disease cluster (e.g., *PRPF38B* for angina pectoris and myocardial infarction, Supplementary Data 10).

When comparing genes and pathways crossing different disease clusters, we found several common genes (e.g., *AGAP1, AOAH, C8orf59* for both clusters featured by eye and ear diseases, Fig. 5A), although no common pathways. Further protein–protein interaction (PPI) analysis identified ten MCODE components (e.g., MCODE components featured by pathways of 'Transmembrane signal transduction' and 'Signaling by G protein-coupled receptor (GPCR)', Fig. 5B) shared by disease clusters of eye, ear and skin and genitourinary diseases.

In sensitivity analyses using the disease clusters and their component medical conditions that can be validated in the UK cohort alone (Table 2 and Supplementary Data 11), we found some identical genetic determinants. Specifically, for the cluster predominated by eye diseases, 11 independent SNPs (e.g., rs578045293, rs574810100, rs192296901) and 115 mapped genes (e.g., *DMRTA2, FAF1, CDKN2C*)

were identified in both genetic analyses. For the cluster predominated by ear diseases, five independent SNPs (e.g., rs113248357, rs193072412, rs556283981), 183 mapped genes (e.g., *RHCE, TMEM57, LDLRAP1*), and two biological pathways (i.e., R-HSA-3700989 Transcriptional Regulation by TP53, R-HSA-1475029 Reversible hydration of carbon dioxide) were identified in both.

Furthermore, comparing the mapped genes for each disease cluster among individuals with anxiety/stress-related disorders (n = 27,781) to those obtained among individuals without anxiety/ stress-related disorders (n = 452,148, Supplementary Data 12), we found only a small proportion of overlapping genes (i.e., 5 and 2 genes for clusters predominated by ear diseases and cardiovascular disease, respectively), indicating that few identified genetic hits were driven by the disease clusters only.

## Subgroup analyses
We found largely similar disease clusters when using the entire Swedish cohort, although the clusters dominated by eye diseases, ear diseases, and cardiovascular disease were merged into one cluster (Supplementary Fig. 4 and Supplementary Table 3). When exploring the disease clusters following anxiety and stress-related disorders separately, we found an additional disease cluster predominated by cerebrovascular diseases for both disorders and another cluster predominated by musculoskeletal diseases for stress-related disorders (Supplementary Figs. 5, 6). We obtained similar disease clusters among females as in the main analysis, although two clusters predominated by cerebrovascular diseases and digestive diseases were also identified (Supplementary Fig. 7). The cluster predominated by ear diseases was

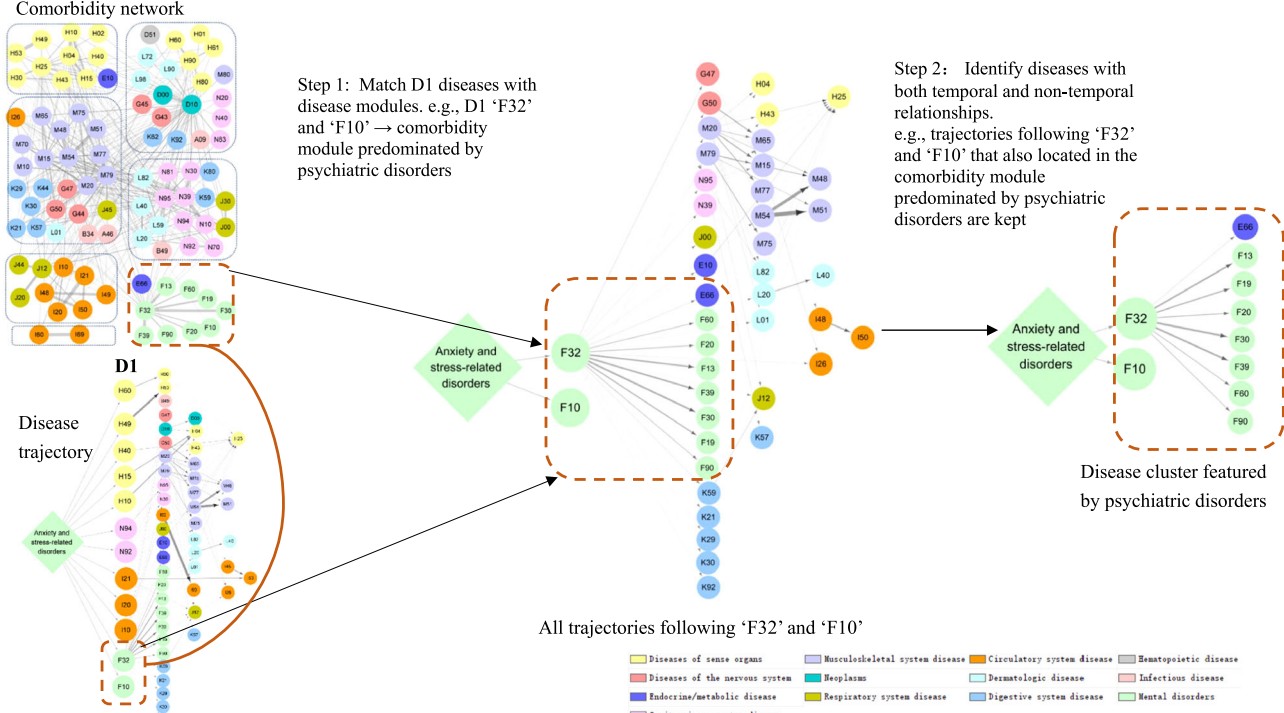

**Fig. 3 | Identification of disease cluster featured by psychiatric disorders by combination of disease trajectory and comorbidity network.** Each node represents a medical condition, and the combined ICD-10 codes are shown within the circle, while the color of the node indicates the category of the corresponding medical condition. Definition of combined ICD-10 codes can be found in Supplementary Data 1. D1 disease 1.

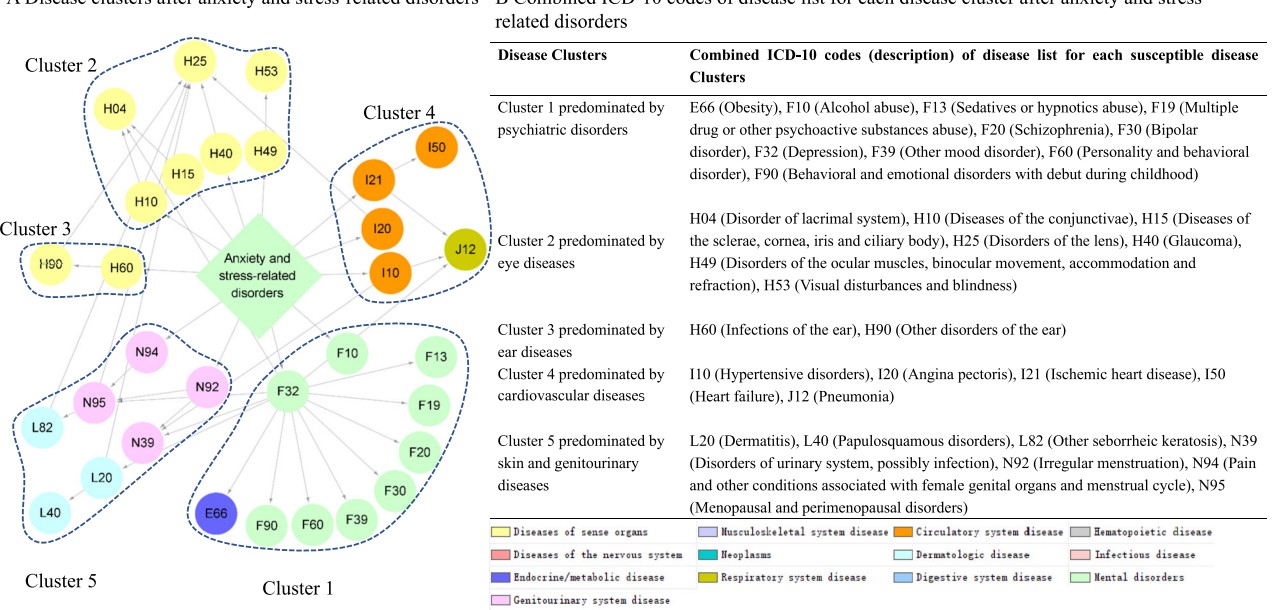

**Fig. 4 | Disease clusters and disease list among individuals with anxiety and stress-related disorders. A** Disease clusters after anxiety and stress-related disorders. Each node represents a medical condition, and the combined ICD-10 codes are shown within the circle, while the color of the node indicates the category of the corresponding medical condition. One disease cluster was defined as D1 and its subsequent diseases from the disease trajectory in Fig. 2A, which belonged the same module in the comorbidity network in Fig. 2B. The network contains five disease clusters, and nodes belonging to the same disease cluster were grouped together using blue dashes. **B** Combined ICD-10 codes of disease list for each disease cluster after anxiety and stress-related disorders. This table lists combined ICD-10 codes for each disease cluster. Definition of combined ICD-10 codes can be found in Supplementary Data 1.

**Table 2 | Genetic determinants for five disease clusters associated with anxiety and stress-related disorders**

| Disease clusters | Number of independent significant SNPs[a] (all candidate SNPs[b]) [Top 5 independent significant SNPs] | Number of mapped genes [Top 10 mapped genes][a] | Top 5 enrichment pathways (IDs and their names)[a] | Involved biological process[c] |
|---|---|---|---|---|
| Cluster 1 predominated by psychiatric disorders | 3 (43) [rs2834342, rs587598040, rs190986212] | 20 [AP000304.12, ATP5O, MRPS6, SLC5A3, KCNE2, SMIM11, C21orf140, AP000322.53, RCAN1, CLIC6...] | GO:0098660 inorganic ion transmembrane transport, GO:0006812 monoatomic cation transport | Molecular transport |
| Cluster 2 predominated by eye diseases | 33 (680) [**rs578045293, rs574810100, rs192296901**, rs185666753, rs182441312...] | 362 [ZFAND1, CHMP4C, SNX16, **DMRTA2, FAF1, CDKN2C, C1orf185, C1orf123, MAGOH,** STXBP6...] | WP5224 2q37 copy number variation syndrome, GO:0070848 response to growth factor, hsa04550 Signaling pathways regulating pluripotency of stem cells, GO:0045165 cell fate commitment, GO:0030900 forebrain development | Signaling by GPCR [e.g., R-HSA-372790 Signaling by GPCR], Brain development [e.g., GO:0030900 forebrain development, GO:0048854 brain morphogenesis], Metabolic process [e.g., hsa00910 Nitrogen metabolism], Transmembrane transport [e.g., R-HSA-199991 Membrane Trafficking] |
| Cluster 3 predominated by ear diseases | 40 (373) [rs567859745, rs559292142, rs192939845, **rs113248357**, rs534986878...] | 440 [C10orf88, PSTK, TEX36, EDRF1, KIAA0895, ANLN, AOAH, ELMO1, **RHCE, TMEM57**...] | GO:0048545 response to steroid hormone, GO:0006066 alcohol metabolic process, GO:0051932 synaptic transmission GABAergic, M3468 NABA ECM REGULATORS, WP696 Benzo(a)pyrene metabolism | Response to stress [i.e., GO:0006979 response to oxidative stress], Alcohol metabolism [e.g., GO:0006066 alcohol metabolic process, GO:0097305 response to alcohol, GO:0034311 diol metabolic process], Molecular transport [i.e., GO:0034220 monoatomic ion transmembrane transport], Metabolic process [e.g., WP696 Benzo(a)pyrene metabolism, GO:0006066 alcohol metabolic process] |
| Cluster 4 predominated by cardiovascular diseases | 4 (41) [rs146697298, rs185984329, rs187327166, rs534240087] | 30 [EFNA5, VAV3, SLC25A24, FAM102B, HENMT1, PRPF38B, FNDC7, KCNA2, LRIF1, DRAM2...] | M5884 NABA CORE MATRISOME, R-HSA-1474244 Extracellular matrix organization, WP411 mRNA processing, hsa05168 Herpes simplex virus 1 infection | NA |
| Cluster 5 predominated by skin and genitourinary diseases | 16 (638) [rs142495798, rs138931818, rs566677164, rs183299296, rs575990437...] | 229 [BCHE, ZBBX, OR6S1, AE000662.92, DAD1, ABHD4, OR6J1, OXA1L, LRP10, REM2...] | GO:0097484 dendrite extension, GO:0034645 Cellular macromolecule biosynthetic process, GO:0050999 regulation of nitric-oxide synthase activity, GO:0045862 positive regulation of proteolysis, R-HSA-442660 Na+/Cl-dependent neurotransmitter transporters | Signaling by GPCR [i.e., WP247 Small ligand GPCRs], Regulation of structure and function of protein [e.g., GO:0045862 positive regulation of proteolysis, GO:0016567 protein ubiquitination, GO:0031647 regulation of protein stability, GO:0018230 peptidyl-L-cysteine S-palmitoylation, R-HSA-9694635 Translation of Structural Proteins] |

In this table, independent SNPs, genes, and enrichment pathways were ordered by p-value from each relevant analysis. SNPs, genes, and enrichment pathways with bold were validated in the sensitivity analysis based on the validated disease clusters in the UK cohort.

SNPs single nucleotide polymorphisms, GWAS genome-wide association study, NA not applicable.

[a]Full lists and information of SNPs, genes, and enrichment pathways were listed in Supplementary Tables 8–10.

[b]All candidate SNPs were in LD with the independent significant SNP at $r^2 \geq 0.6$.

[c]Involved biological processes were summarized based on the category of enrichment pathways.

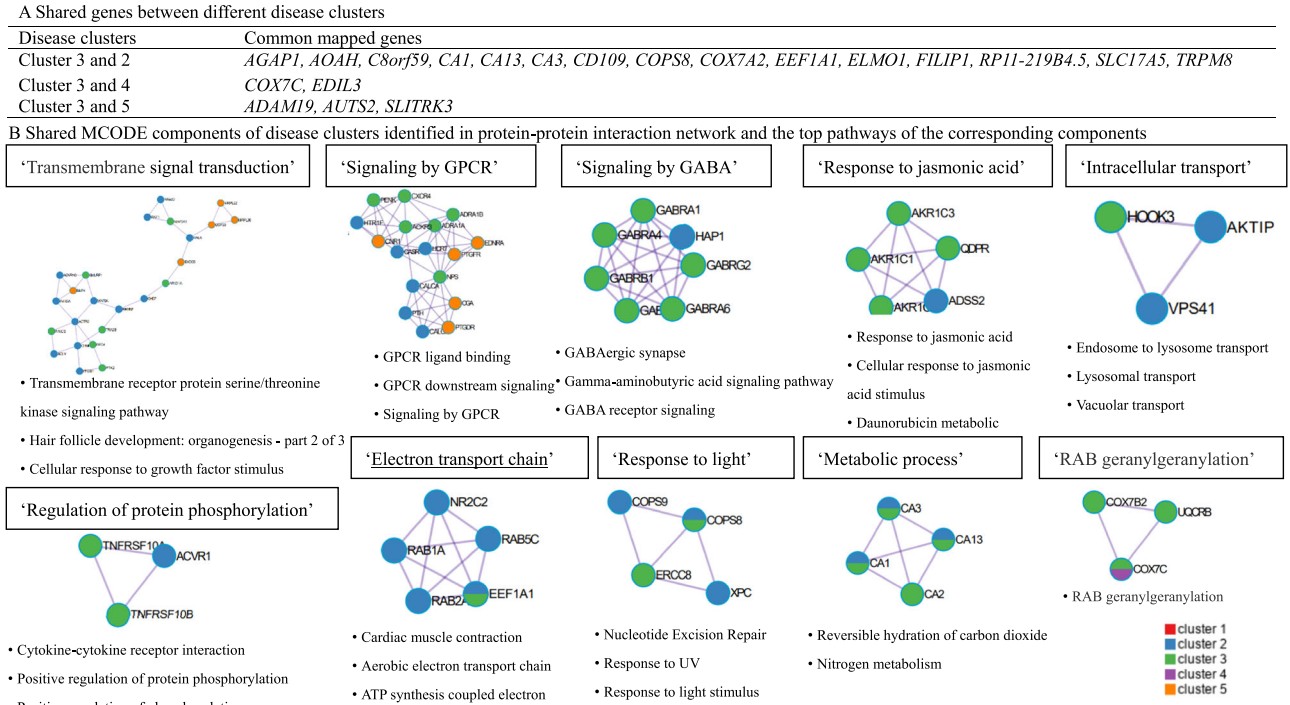

**A** Shared genes between different disease clusters

| Disease clusters | Common mapped genes |
|---|---|
| Cluster 3 and 2 | *AGAP1, AOAH, C8orf59, CA1, CA13, CA3, CD109, COPS8, COX7A2, EEF1A1, ELMO1, FILIP1, RP11-219B4.5, SLC17A5, TRPM8* |
| Cluster 3 and 4 | *COX7C, EDIL3* |
| Cluster 3 and 5 | *ADAM19, AUTS2, SLITRK3* |

**B** Shared MCODE components of disease clusters identified in protein–protein interaction network and the top pathways of the corresponding components

**Fig. 5 | Genetic overlap between five disease clusters associated with anxiety and stress-related disorders. A** Common gene between different disease clusters. **B** Shared MCODE components of disease clusters identified in protein–protein interaction network and the top pathways of the corresponding components, where each node represents a protein with a pie chart encoding its origin. SNPs: single nucleotide polymorphisms. Cluster 1 predominated by psychiatric disorders, Cluster 2 predominated by eye diseases, Cluster 3 predominated by ear diseases, Cluster 4 predominated by cardiovascular diseases, Cluster 5 predominated by skin and genitourinary diseases. GWAS genome-wide association study, GPCR G Protein Coupled Receptors, MCODE Molecular Complex Detection algorithm.

not identified but a disease cluster predominated by musculoskeletal diseases was noted among males (Supplementary Fig. 8).

## Discussion

Leveraging the nationwide health registers in Sweden and the large community-based UK Biobank, our study, revealed the most comprehensive picture of subsequent disease clusters following a diagnosis of anxiety and stress-related disorders. The five distinct disease clusters, featured by psychiatric disorders, eye diseases, ear diseases, cardiovascular diseases, and skin and genitourinary diseases, were discovered in the Swedish cohort and validated in the UK cohort. Furthermore, based on individual-level genotyping data in the UK cohort, we identified several distinct genetic determinants for the five disease clusters, as well as genetic components involved in the GPCR signaling pathway that were shared between multiple disease clusters. With novel attempts to conceptualize associated disease clusters, these findings shed light on the biological basis, both commonly and specifically, towards further diverse health consequences after a diagnosis of anxiety and stress-related disorders, which could aid mechanistic explorations, and facilitate risk surveillance (e.g., precise risk assessment) and management (e.g., development of targeted interventions) for health decline prevention among patients with newly diagnosed anxiety and stress-related disorders.

Over the past decade, accumulated evidence suggests a positive link between anxiety or stress-related disorders and a number of medical conditions. Previous studies have, however often relied on small samples[24], with incomplete follow-up[25], and focused on a single outcome/disease[11,26]. Our study, therefore complements the knowledge gaps through a data-driven approach of including virtually all medical conditions after the diagnosis of anxiety and stress-related disorders. As a result, we managed to identify five key clusters of disease associated with a prior diagnosis of anxiety or stress-related disorders (with component diseases in the same system or across

different systems), considering the temporal order and high intrinsic connectivity between diseases and with validation across two populations. Although evidence on the risk of these disease clusters after anxiety and stress-related disorders was scarce, our findings gain support from previous studies reporting associations between anxiety or stress-related disorders and individual diseases. For instance, anxiety and stress-related disorders have previously been reported to co-occur with other psychiatric disorders (e.g., major depressive episode, bipolar disorder, and alcohol dependence)[27]. Additionally, a population-based cohort study covering 5.9 million people in Denmark reported increased risks of 31 somatic medical conditions, including cardiovascular disease, vision problems, and hearing problems, following a prior diagnosis of neurotic disorders (including anxiety and stress-related disorders)[28]. In our previous population-based cohort study in Sweden, we also noted risk increases of 16 specific cardiovascular diseases among patients with stress-related disorders[12]. In addition, several diseases in disease clusters predominated by skin and genitourinary diseases have been reported to be associated with anxiety or stress-related disorders, such as urinary infection[29], irregular menstruation[30], and premenstrual syndrome[31]. We found largely similar disease clusters following a diagnosis of anxiety and stress-related disorders, with an additional cluster of musculoskeletal diseases noted among patients with stress-related disorders. Several studies have reported similar associations between post-traumatic stress disorder (PTSD) and arthritis, although using self-reported data[32,33]. Furthermore, we found little role of age and sex in the identification of most disease clusters, indicating that the risk of developing these disease clusters is independent of age and sex.

Prior attempts to illustrate disease networks include a recent Danish study based on national data from inpatient and outpatient care, which established a browser presenting the disease trajectories both before and after a target disease of interest[34]. Using this browser, we found that some key diseases in our study, such as cardiovascular

diseases identified subsequent to stress-related disorders, were listed prior to stress-related disorders. However, as the comparability of these two studies is limited (due to the different research purposes and study designs), the inconsistent results do not necessarily invalidate each other.

With the notion that diseases with high levels of connectivity (i.e., in a disease cluster) may share common pathological mechanisms with possibly the same affected genes and biological pathways, we consider it is reasonable to focus on disease clusters, instead of each individual disease, for the purpose of genetic determinant identification and for the future development of disease prevention strategies. Despite the lack of comparable results from studies of similar design, the findings of our cluster-specific genetic analyses were in line with prior studies. For instance, some mapped genes for susceptibility to the disease cluster featured by psychiatric disorders (e.g., *AMOTL1, CWC15, KDM4D*) were reported to be associated with mood disorders[35] as well as attention deficit hyperactivity disorder (ADHD) and conduct disorder[36]. *IQCB1* and *GOLGB1*, the genes identified for the disease cluster dominated by eye diseases, have been associated with corneal resistance factor (a measure of the biomechanical properties of the cornea)[37], while *CHDH* and *FILIP1L* have been associated with ocular axial length[38]. Additionally, some mapped genes for the disease cluster dominated by cardiovascular disease have been demonstrated as risk genes for diastolic blood pressure (*COL23A1* and *PHYKPL*)[39] or stroke (*COL23A1, PHYKPL,* and *ADAMTS2*)[40,41]. Nevertheless, we found several risk genes (e.g., *ATP5O, POLQ, RHCE*) and biological pathways not discussed in the existing literature, mainly involved in the regulation of molecular transport, cellular metabolic processes, and the structure and function of proteins. Particularly, it is also notable that, except for genes and pathways that were linked to a specific disease cluster, we identified the biological components related to the signaling of GPCR that may contribute to the development of multiple disease clusters (i.e., clusters featured by eye, ear, and skin and genitourinary diseases). GPCR signaling has been widely reported to play a role in responses to stress[42], inflammatory response[43], and development and drug targets for multiple diseases[44] (e.g., dry eye disease[45], allergic conjunctivitis[46], and urinary tract infection[47]), which may indicate the key shared pathways and mechanisms linking anxiety and stress-related disorders to subsequent sequelae. Collectively, if verified, the findings of our study might provide additional insights into why patients diagnosed with anxiety and stress-related disorders face a general health decline, with large variations in developed disease outcomes. Regardless, our measure of susceptibility score to a disease cluster rather than to a single disease, might limit the comparison of findings between the present study and previous studies.

Our efforts to identify of disease clusters and their genetic determinants were conceptual and were largely based on accumulating evidence of the existence of disease networks and their shared biological mechanisms[20,21,48], with the potential to aid in the development of cost-effective health promotion strategies for this vulnerable population. For instance, medications indicated for the genes/pathways within each disease cluster could be further tested for effectiveness in reducing risks of further disease development among individuals with anxiety and stress-related disorders. Other major strengths of our study include the inclusion of two large population- and community-based cohorts with long and complete follow-up data collected prospectively and independently, which largely minimized information and selection biases, and enabled the ascertainment of disease clusters using data from two distinct populations. Furthermore, the combination of disease trajectory and comorbidity network analyses to ascertain disease clusters enhanced the reliability of the connectivity and temporal order between disease pairs in each disease cluster. Last, the availability of enriched phenotypic and genotypic data, together with the application of comprehensive analytic strategies, including PheWAS, disease network analysis, and genetic analysis,

for the first time, led to a comprehensive illustration of health consequences in relation to anxiety and stress-related disorders from phenotypic to genetic levels. This analytic strategy could be applied as a pipeline for studying comorbidities of other phenotypes.

Several limitations should be acknowledged. First, given the lack of complete primary care data in the Swedish Patient Register and the UK Biobank, as well as the lack of outpatient care data in the UK Biobank, we might have underestimated the number of patients with anxiety and stress-related disorders as well as the number of studied medical conditions, primarily the milder forms of these diseases. Therefore, disease cluster identification based on a more comprehensive data source to validate the findings of the present study is warranted. Second, although we excluded patients with a history of other psychiatric disorders and severe somatic diseases and started the follow-up from six months after the index date, we cannot rule out the possibility that some pre-existing diseases other than anxiety and stress-related disorders might have contributed to the identified disease clusters. Third, the identification of disease clusters relied on the results of association analyses (i.e., PheWAS analysis). Although confirmed by using two methods and validated in the UK cohort, the lack of data on important confounders in the health register data (e.g., lifestyle and environmental factors) can raise the concern of residual confounding. This also applies to the noted negative association of anxiety/stress-related disorders with varicose veins of lower extremities. With few supportive data from existing literature, such a finding needs to be validated in future studies. Last, our findings may not be generalized to other populations with non-European ancestry or different healthcare systems than in Sweden and the UK.

In conclusion, based on detailed phenotypic and genetic analyses of two large-scale cohorts, we identified five distinct disease clusters subsequent to an inpatient/outpatient diagnosis of anxiety and stress-related disorders, featured by other psychiatric disorders, eye diseases, ear diseases, cardiovascular diseases, and skin and genitourinary diseases as predominant diseases in each cluster. We further identified a list of genetic variants and biological pathways linking anxiety and stress-related disorders, specifically or commonly, to those identified disease clusters, contributing to a better understanding of the underlying mechanisms.

## Methods
### Study design
The analytic process included two parts, namely phenotypic and genetic analyses (Fig. 1). In the phenotypic analysis, we undertook a phenome-wide association study (PheWAS), followed by both comorbidity network analysis and disease trajectory analysis to determine robust disease clusters (i.e., associated diseases with both temporal and non-temporal relationships) following a diagnosis of anxiety or stress-related disorders in the Swedish cohort (i.e., the exploratory dataset). To validate identified disease clusters and the diseases they entailed, we performed similar analyses in the UK cohort of similar study designs (i.e., the validation dataset). Regarding the genetic analyses based on individual-level genotyping data of the UK cohort, we first calculated a cluster-specific susceptibility score, which was designated as a quantitative index of individuals' susceptibility to a specific disease cluster, and then performed GWAS analysis, gene mapping, and enrichment analysis to identify risk genes and biological pathways that may count for the pathogenesis of such a disease cluster after anxiety or stress-related disorders.

### Swedish cohort
The Swedish Patient Register includes nearly complete health records of inpatient care since 1987 and outpatient specialist care since 2001 in Sweden[49]. By cross-linkage to the Total Population Register using the unique Swedish personal identification numbers, we included all Swedish-born individuals residing from 2001 to 2016 in Sweden and

excluded those with any pre-existing psychiatric disorders or history of severe somatic diseases at the time of diagnosis determined by the Charlson Comorbidity Index before 2001[50], leading to a study population of 8,456,485 (Supplementary Fig. 1). We focused on Swedish-born individuals in the present study to reduce the heterogeneity in genetic background as well as other sociodemographic factors, including differential health-seeking behaviors. Among these, we identified all individuals who received a first primary diagnosis in specialized care of anxiety or stress-related disorders from 2001 to 2016 ($N = 212,767$, 63.3% with anxiety disorder), and a set of ten unaffected individuals randomly selected from the study base per exposed patient, individually matched on sex and birth year using incidence density sampling ($N = 2,127,670$), without history of other psychiatric disorders and severe somatic diseases. The diagnostic date of anxiety or stress-related disorders was used as the index date for the start of follow-up of both the exposed patients and their matched unaffected individuals.

### Follow-up

To minimize the concern of reverse causality, we followed all participants of the Swedish cohort for all medical conditions from 6 months after the index date until death, first diagnosis of anxiety or stress-related disorders (for matched unexposed individuals), emigration, or the end of the study period (i.e., 31 December 2016), whichever occurred first.

### Ascertainment of anxiety and stress-related disorders and subsequent medical conditions

In the Swedish cohort, we defined anxiety or stress-related disorders as any first specialist care diagnosis in an inpatient or outpatient hospital visit, where these disorders were identified as the primary discharge diagnosis, according to the Swedish Patient Register, using the 10th Swedish revision of the International Classification of Diseases (ICD-10) codes (anxiety: F40 and F41, stress-related disorder: F43) (Supplementary Table 1). In the PheWAS, medical conditions refer to any disease or health outcomes recorded in the Patient Register comprising inpatient and outpatient diagnoses. We ascertained medical conditions through the primary diagnosis from the Patient Register, using the 3-digit ICD-10 codes (A00 to N99) (Supplementary Table 1). The diagnostic codes for most common diseases in the Patient Register have been validated, showing a satisfactory accuracy with positive predicted values [PPV] of 85–95% for most common diseases[49], 81% for social anxiety disorder[51], and 75–90% for PTSD[52]. We obtained the highest level of education and income at the year of index date from the Swedish Longitudinal Integration Database for Health Insurance and Labor Market[53].

### UK cohort

The validation dataset (UK cohort) was constructed based on the UK Biobank, using a similar design (Supplementary Fig. 2). The UK Biobank (UKB) is a community-based cohort study that enrolled half a million participants aged 40–69 at recruitment between 2006 and 2010 across England, Scotland, and Wales. Details of the study design are described elsewhere[54]. The inpatient hospital data, obtained from the Hospital Episode Statistics database, the Scottish Morbidity Record, and the Patient Episode Database, cover all UK Biobank participants since 1997[55]. The primary care data, provided by various general practitioner computer system suppliers, cover ~45% of participants since 1985[55]. We first excluded individuals who had withdrawn from the UK Biobank ($n = 108$) or had conflicting information ($n = 1$), leaving 502,398 eligible participants (Supplementary Fig. 2). Among these, we constructed a matched cohort, including patients with newly inpatient/primary care diagnosed anxiety or stress-related disorders between January 1, 1997 and December 31, 2019 ($N = 23,365$) who had no history of severe somatic diseases or other psychiatric disorders, and up to ten unaffected individuals for each patient who were randomly selected and individually matched by sex and year of birth using incidence density sampling ($N = 233,596$). The diagnostic date of anxiety or stress-related disorders was used as the index date for the start of follow up of the exposed and matched unaffected individuals.

We followed all participants of the UK cohort for all medical conditions from 6 months after the index date until death, first diagnosis of anxiety or stress-related disorders (for matched unexposed individuals), loss to follow-up[56], or the end of the study period (i.e., 31 December 2019), whichever occurred first.

In the UK cohort, we defined a new diagnosis of anxiety or stress-related disorder as a first primary diagnosis based on the inpatient and primary care data, using the ICD-10 codes for the inpatient hospital data (Supplementary Data 1) and the version 2 and version 3 read codes (i.e., Read v2 and Read v3) for the primary care data (Supplementary Table 4). Medical conditions were ascertained from the primary and secondary diagnoses through the inpatient hospital data (Supplementary Data 1). Information on the highest educational level and Townsend Deprivation Index (proxy for socioeconomic status, with a higher index score indicating a higher degree of deprivation)[57] were collected at recruitment through questionnaires.

### Statistical analyses

The age distribution differed between the Swedish cohort and the UK cohort (median age at diagnosis 32 versus 52). To facilitate validation between the two cohorts, we selected a sub-cohort of the Swedish cohort, namely participants with age >second tertile (median age at index date = 51, $N = 70,026$, Table 1), and used this sub-cohort as the exploration dataset throughout the main analyses. A sensitivity analysis was performed using the entire Swedish cohort with all age groups.

### Exploration of associated disease clusters in the Swedish cohort

In the exploration dataset (Swedish cohort), we identified a total of 454 medical conditions diagnosed six months after the diagnosis of anxiety or stress-related disorders. To ensure statistical power, we included only medical conditions with a prevalence ≥0.5% among patients with anxiety or stress-related disorders. We performed a PheWAS to investigate the associations between anxiety or stress-related disorders and each medical condition, using Cox regression models stratified by matching variables (i.e., sex and birth year) with adjustment for highest education and income. Individuals with a prior diagnosis of the studied medical condition were excluded, when estimating the association with each medical condition. Only medical conditions with statistically significant positive associations, after adjusting for multiple testing (hazard ratio [HR] > 1, and false discovery rate [FDR] adjusted $p$ value [i.e., $q$ value] <0.05), were included in the following analyses.

Among the identified medical conditions from the PheWAS, we constructed all possible disease pairs as disease 1 (D1) and disease 2 (D2) pairs and only analyzed disease pairs that co-occurred with a prevalence ≥0.25% among patients with anxiety or stress-related disorders. To ensure comorbidity strength, we calculated the relative risk (RR) and Pearson's correlation ($\Phi$-correlation) for each disease pair. For each disease pair, a sub-cohort was formed through excluding patients with a history of D1 and D2 before their index date (i.e., the diagnosis date of anxiety or stress-related disorders). The formulas for RR and $\Phi$-correlation were calculated using the following formulas:

$$RR_{ij} = \frac{C_{ij}N_{ij}}{C_i C_j}$$

$$\Phi_{ij} = \frac{C_{ij}N_{ij} - C_i C_j}{\sqrt{C_i C_j (N_{ij} - C_i)(N_{ij} - C_j)}}$$

Where $C_{ij}$ is the number of patients affected by both D1 and D2, and $N_{ij}$ is the number of individuals in the sub-cohort, while $C_i$ and $C_j$ are the number of patients affected by D1 and D2 respectively. For both RR and $\Phi$-correlation measures, the significance of RR = 0 and $\Phi = 0$ can be both determined using z-test (given large sample size in our study). The corresponding z-score for RR and $\Phi$-correlation were calculated using the following formula[21,58]:

$$z_{ij}^{\mathrm{RR}} = \frac{\ln(RR_{ij})}{\sqrt{\frac{1}{C_{ij}} - \frac{1}{N_{ij}} + \frac{1}{C_i C_j / N_{ij}} - \frac{1}{N_{ij}}}}$$

$$z_{ij}^{\Phi} = \frac{\Phi_{ij}\sqrt{\max(C_{ij}, C_j) - 2}}{\sqrt{1 - \Phi_{ij}^2}}$$

*P* values were then calculated using the z-score and adjusted for the issue of multiple testing. Only disease pairs with strong comorbidity strength (i.e., RR > 1, $\Phi$-correlation > 0, and *q* value < 0.05) were included in the comorbidity network and disease trajectory analyses.

In the comorbidity network analysis, we used logistic regression to determine the magnitude of association between the disease pairs with strong comorbidity strength (i.e., significant non-temporal relationship). Disease pairs with confirmed positive association (i.e., odds ratio [OR] > 1 and *q* value < 0.05) were selected to construct a comorbidity network. The comorbidity network was then subdivided into different comorbidity modules with high intrinsic connectivity determined by the Louvain clustering algorithm[59]. For disease trajectory analysis, binomial tests were used to assess the temporal direction (i.e., D1 → D2 or D2 → D1 among D1D2 pairs) among disease pairs with strong comorbidity strength (i.e., significant temporal relationship). For each disease pair with a determined temporal order, we constructed a nested case-control dataset in the sub-cohort, by considering D2 as outcome and D1 as exposure. For each patient with D2, at most two controls were matched by sex and birth year using intensity density sampling. to confirm the magnitude of the association between the disease pair, we then used conditional logistic regression by adjusting for education level and Townsend Deprivation Index. We then included the disease pairs with positive associations (OR > 1 and *q* value < 0.05) to construct the disease trajectory.

As disease trajectory analysis is designed to visualize sequential disease progression while comorbidity network analysis captures disease groups with high intrinsic connectivity, the combined use of those two data-driven approaches can theoretically lead to the identification of more reliable disease clusters (i.e., groups of diseases with both temporal and non-temporal relationships). Thus, based on results from the aforementioned disease trajectory and comorbidity network analyses, we defined disease clusters as the first layer diseases (D1) and their subsequent diseases in a disease trajectory that were also located within the same comorbidity module (Fig. 3). For example, in the disease trajectories, out of all diseases in the first layer, "F32" and "F10" were located in one comorbidity module (i.e., the module predominated by psychiatric disorders) derived from the comorbidity network. We then found the following diseases of "F32" and "F10" in the trajectories which were also in such a comorbidity module to constitute a disease cluster (i.e., the disease cluster featured by psychiatric disorders including "E66", "F10", "F13", "F19", "F20", "F30", "F32", "F39", "F60", and "F90").

### Validation of associated disease clusters in the UK cohort
To validate the identified disease clusters in the UK cohort, we used Cox models to assess the associations between anxiety and stress-related disorders and each medical condition of the identified disease cluster, comparing affected patients to their matched unaffected individuals. The models were stratified by matching variables (i.e., sex

and birth year) and adjusted for highest educational level and Townsend Deprivation Index. Only medical conditions with statistically significant positive associations were included in the comorbidity network analysis (same as described above) to construct disease clusters in the UK cohort. Trajectory analysis was not performed in the UK cohort due to the lack of complete primary care and outpatient data.

### Genetic determinants of associated disease clusters using data from the UK Biobank
In the UK cohort, a cluster-specific "susceptibility score" was calculated to quantify the subsequent risk of each disease cluster for each patient with anxiety or stress-related disorder. The susceptibility score was defined as an individual person's number of diagnosed diseases included in each disease cluster, according to the inpatient hospital data.

The quality control contains two parts. For quality control on individuals from the UK Biobank, we removed individuals with non-European ancestry, inconsistent sex, or sex chromosome aneuploidy. For quality control on individual level of genetic data, we first removed SNPs with imputation quality score <0.8, minor allele frequency < 0.001, or deviations from Hardy-Weinberg equilibrium ($p < 1 \times 10^{-10}$)[60]. We removed SNPs in the extended major histocompatibility complex (MHC) region (chr6: 25–34 Mb), considering the long-range linkage disequilibrium (LD) and special genetic architecture in this region. After standard GWAS quality control on the individual-level genotyping data[60], we included 22,781 patients (out of the 23,365 patients) and 13,225,429 variants for further analysis.

To assess the association between SNPs and five susceptibility scores (as a continuous variable) respectively, we used mixed linear model (MLM)-based models for GWAS analysis, adjusted for sex, birth year, genotyping array, and the first ten principal components[61]. Independent significant SNPs with $p < 5 \times 10^{-8}$ were identified for each genomic locus. The inflation of GWAS analyses was tested by LD score regression[62]. Together with surrounding genomic loci that were identified based on LD structure at $r^2 \geq 0.6$, all the SNPs were further mapped to identify potential genes using the web-based platform FUMA (http://fuma.ctglab.nl/)[63]. The strategies for gene mapping included positional mapping, expression quantitative trait loci mapping based on the GTEx v8 Project[64], and Chromatin interaction mapping[65,66]. Further, these mapped genes were included in the gene-set enrichment analyses based on Metascape (https://metascape.org/) to identify underlying biological pathways for each disease cluster with the following ontology sources: KEGG Pathway, GO Biological Processes, Reactome Gene Sets, Canonical Pathways, CORUM, WikiPathways, and PANTHER Pathway[67]. To further investigate genetic overlap across disease clusters, the abovementioned mapping genes were included in the PPI network enrichment analysis to identify protein network components using the Molecular Complex Detection (MCODE) algorithm based on Metascape[68] based on the following genomic interaction databases: STRING[69], BioGrid[70], OmniPath[71], and InWeb_IM[72]. To test whether these identified cluster-specific genes are associated with any of the individual diseases in each disease cluster, we first conducted hypergeometric tests using the function of "GENE2FUNC" in the website platform 'FUMA'. As an alternative approach serving a similar purpose, we searched GeneCards, a gene database providing all annotated and predicted human genes (https://www.genecards.org/), to identify traits that have been previously associated with the top 20 identified genes of each disease cluster.

In a sensitivity analysis, we repeated the genetic analysis merely for the disease clusters (and their components) that can be validated from the UK cohort. Additionally, to test whether the identified genes were primarily driven by the studied disease clusters, independent of the prior anxiety/stress-related disorders, we conducted additional GWAS analyses for those disease clusters among individuals without anxiety/stress-related disorders ($n = 452{,}148$).

## Subgroup analyses

As we used the advanced age group (i.e., >second tertile) of the Swedish cohort in the main analysis, we repeated the phenotypic analyses in the entire Swedish cohort ($N = 216,727$). To assess whether the results would differ between anxiety and stress-related disorders, we constructed two independent matched cohorts for anxiety and stress-related disorders, separately, and repeated the main analyses to identify disease clusters associated with anxiety disorder and stress-related disorder exclusively. To explore the role of sex in disease clusters, we performed analyses separately for males and females.

The phenotypic analyses were conducted using SAS 9.4 (SAS Institute), R (Version 4.0.2), Python (Version 3.8), and Cytoscape (Version 3.8.0). PLINK (Version 1.9) and GCTA (Version 1.24) were used for genetic analyses. For multiple testing, a $q$ value < 0.05 was considered statistically significant. This study was approved by the Ethical Vetting Board in Stockholm, Sweden (DNRs 2012/1814-31/4 and 2015/1062-32), the NHS National Research Ethics Service (16/NW/0274), and the Biomedical Research Ethics Committee of West China Hospital (2019-1171). The requirement of informed consent for Swedish participants is waived in register-based studies in Sweden, and all the participants in the UK Biobank provided written informed consent before data collection.

## Reporting summary

Further information on research design is available in the Nature Portfolio Reporting Summary linked to this article.

## Data availability

The raw data from Swedish registers are protected and not available due to Swedish law. The raw data from the UK Biobank (http://www.ukbiobank.ac.uk/) are available to all researchers upon making an application. Part of this research was conducted using the UK Biobank Resource under Application 54803. Other data or platforms are available to all researchers: FUMA (http://fuma.ctglab.nl/), Metascape (https://metascape.org/), GeneCards (https://www.genecards.org/).

## Code availability

As the codes are highly specific to our curated database and may not be universally applicable to others, all codes associated with the current submission are available and can be requested by contacting the corresponding authors.

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

## Acknowledgements
We thank the team members and colleagues involved in the West China Biomedical Big Data Center- UK Biobank project for their support. This work is supported by the National Natural Science Foundation of China (No. 81971262 to H.S.), 1.3.5 project for disciplines of excellence, West China Hospital, Sichuan University (No. ZYYC21005 to H.S.), EU Horizon2020 Research and Innovation Action Grant (847776 to U.V. and F.F.), Consolidator grant from the European Research Council (726413 to U.V.), the Fundamental Research Funds for Central Universities (No. 20826041F4144 to X.H.), the Outstanding Clinical Discipline Project of Shanghai Pudong (No.PWYgy2021-02 to Q.S.) and the Fundamental Research Funds for the Central Universities (to Q.S.). This research has been conducted using the UK Biobank Resource under Application 54803. This work uses data provided by patients and collected by the NHS as part of their care and support. This research used data assets made available by National Safe Haven as part of the Data and Connectivity National Core Study, led by Health Data Research UK in partnership with the Office for National Statistics and funded by UK Research and Innovation (grant ref: MC_PC_20029 and MC_PC_20058).

## Author contributions
H.S. and U.A.V. were responsible for the study concept and design. X.H., Q.S., H.Y., W.C., Y.Z., Y.Q. and W.Y. did the data and project management. X.H., Q.S. and C.H. did the data cleaning and analysis. X.H., Q.S., C.S., H.S., U.A.V. and F.F. interpreted the data. X.H., Q.S., U.A.V., F.F. and H.S. drafted the manuscript. All authors approved the final manuscript as submitted and agree to be accountable for all aspects of the work. The corresponding author attests that all listed authors meet authorship criteria and that no others meeting the criteria have been omitted.

## Competing interests
The authors declare no competing interests.
