## [Peer Review File · Nature Communications]

Disease clusters subsequent to anxiety and stress-related disorders and their genetic determinantsREVIEWER COMMENTS

Reviewer #1 (Remarks to the Author):

Use of Swedish patient register with a well-balanced cohort selection of case matching by birth year and sex. Most GWAS maximize sample count above all else, this is a welcome change. Design is interesting and novel. Validation in UKBB is even better. Still UKBB is a bit peculiar in its SES skewing (perhaps not as bad since TDI and education were adjusted for?), are there other national biobanks that could be included for additional validation (US Million Veteran Program, AllofUs, FinnGen, Estonia etc)?

It's nice that the full table of ICD codes is provided in supplement but the definition "we defined anxiety or stress-related disorders as any first primary diagnosis in an inpatient or outpatient hospital visit according to the Swedish Patient Register" is too vague; a more clear designation of what specifically was included to define anxiety and stress-related disorders would be appreciated. Perhaps beyond scope, but focusing on positive associations for the comorbidity network makes me wonder about the negative associations. Resilience factors could be at least as interesting as co-morbid risk factors?

Why only inpatient hospital data in the UKBB data? Outpatient dx codes could also be relevant?

A point of clarity within the genetic determinants method section: the MLM was run against 5 susceptibility scores as a quantitative phenotype? If so, please state clearly.

22,781 seems a small proportion of the UKBB total. Please account more clearly for the participants excluded.

A large number of hits were identified in the GWAS relative to the number of participants (22,781 patients in UKBB if I'm understanding). Current studies of anxiety (Otowa, Purves, Meier, Levey etc) haven't found this many in larger samples. How much of these genetic determinants come from the somatic disease co-morbidities rather than the cluster itself? Meaning, does the cardiovascular cluster 4 discovery mostly show genetic associations with cardiovascular phenotypes? Is the over-representation of loci discovered in cluster 2 and 3 for eye and ear disease surprising in that context? I don't see any report of how inflation was accounted for in the GWAS. Please report λ , intercept and attenuation ratio of ld score regression in main text.

This is a novel and interesting study. I'd like a little more detail on methodology and more clarity on what the results of the genetic determinant analysis means (are these truly co-morbidity markers or would we see these in a univariate analysis of the co-morbid disease trait alone)?

Reviewer #2 (Remarks to the Author):

The present study aimed to determine disease clusters after a diagnosis of anxiety or stress-related disorders and the to identify the genetic basis of these. The study is very comprehensive, both conceptually and methodologically, and it presents with a novel approach towards the study of comorbidity. Nevertheless, several aspects need to be further clarified so as to make the added value of the study more visible.

The authors state that the cluster approach to comorbidity analysis provides a more comprehensive understanding of phenotype heterogeneity in persons with anxiety or stress-related disorders. However, the five clusters identified are truly system/organ-specific and reveal little novelty with respect to the single-disease associations mentioned in previous literature. Besides the limited novelty, these findings contrast with previous literature on patterns of multimorbidity, whereby anxiety was often part of complex multi-system/organ clusters. Related to this, the authors provide little or no comparative insights with previous studies on patterns of multimorbidity in comparable age groups.

Even if the authors mention the limitation linked to the lack of primary care data coverage, they should further elaborate on the sensitivity and specificity of Swedish NPR and UK Biobank data to

capture anxiety or stress-related disorders. I would assume that the biggest part of these diagnoses are made in primary care, and may never reach secondary/tertiary care. This has important implications in terms of sample selection bias that requires further careful and critical elaboration.

It is not entirely clear to me why and how the disease trajectory (temporal relationship) and comorbidity network (non-temporal relationship) analyses are combined. It would be helpful if the authors could provide some real examples in the methods as to how this was performed. Related to this, did the authors not think of other, more straightforward and person-centered clustering approaches such as latent class analysis?

Bearing in mind that anxiety or stress-related disorders are, to a great extent, socially patterned, stratification by socio-economic status would be of high interest and added value. The authors exclude non-Swedish born subjects from their Swedish sample; this decision needs to be better justified, precisely because the immigrant layer is one that strongly intersects with social class.

On a more general level, I would like to inquire the authors concerning the following justification for running one of the objectives in their study. They mention that "the identification of cluster-specific, instead of diseases-specific, genetic variants, have potential to increase the feasibility of developing early intervention for preventing a general health decline among patients with those psychiatric disorders". The authors should further and critically elaborate on their views of how gene-based risk estimates could help in the design of targeted preventive interventions.

Responses to the comments of the referees*:

*All page and line numbers correspond to the page and line numbers of the “**Revised Manuscript_clean**” version.

Comments from Reviewers

Reviewers #1:

Comments:

Use of Swedish patient register with a well-balanced cohort selection of case matching by birth year and sex. Most GWAS maximize sample count above all else, this is a welcome change.

Thank you for the compliment.

1. Design is interesting and novel. Validation in UKBB is even better. Still UKBB is a bit peculiar in it’s SES skewing (perhaps not as bad since TDI and education were adjusted for?), are there other national biobanks that could be included for additional validation (US Million Veteran Program, AllofUs, FinnGen, Estonia etc)?

Authors’ response 1:

Thank you for the positive comment. We agree with the reviewer on the importance of validating findings obtained from one population in an external population. This is why we cross-validated findings between the Swedish registers and the UK Biobank.

However, although we certainly appreciate the reviewer’s suggestion regarding additional validation in other national biobanks, we currently have no access to such biobanks and therefore cannot perform such additional analysis within a reasonable timeframe. However, if the reviewer and editors believe that this is of utmost importance, we are certainly willing to ask for an extension of the revision and apply for data access to some of the suggested biobanks.

We have also addressed this in the Discussion section (Page 16, Lines 306-308) in the revised manuscript:

“Last, our findings cannot be generalized to other populations of non-European ancestry or different health care systems than in Sweden and the UK.”

2. It’s nice that the full table of ICD codes is provided in supplement but the definition “we defined anxiety or stress-related disorders as any first primary diagnosis in an inpatient or outpatient hospital visit according to the Swedish Patient Register” is too vague; a more clear designation of what specifically was included to define anxiety and stress-related disorders would be appreciated.

Authors’ response 2:

Thank you for the comment. We used 3-digit ICD-10 (International Classification of Diseases, 10th revision) codes to identify anxiety (F40-F41) or stress-related disorders (F43) according to the Swedish Patient Register, in line with our previous studies (Song H et al., JAMA, 2018; Chen W et al., BMC Medicine, 2022).

We have now clarified this in the revised manuscript.

“Methods and Materials” section (Page 18, Lines 355-363):

“In the Swedish cohort, we defined anxiety or stress-related disorders as any first specialist care diagnosis in an inpatient or outpatient hospital visit, where these disorders were identified as the primary discharge diagnosis, according to the Swedish Patient Register, using the 10th Swedish revision of the International Classification of Diseases (ICD-10) codes (anxiety: F40 and F41, stress-related disorder: F43) (Supplementary Table 1).”

3. Perhaps beyond scope, but focusing on positive associations for the comorbidity network makes me wonder about the negative associations. Resilience factors could be at least as interesting as co-morbid risk factors?

Authors’ response 3:

Thank you for the interesting thought. In this paper, we focused on observed positive associations, to help identify patients who are potentially at a high risk of developing other diseases following a diagnosis of anxiety or stress-related disorders. However, when checking all assessed associations in the PheWAS analysis of the Swedish cohort, we noted indeed one negative association with statistical significance, namely varicose veins of lower extremities (hazard ratio: 0.93, 95% confidence intervals: 0.89-0.98), when comparing patients with anxiety or stress-related disorders to their matched unexposed individuals. To the best of our knowledge, there is currently little evidence supporting the observed negative association. Therefore, although we have carefully corrected for multiple testing in PheWAS, we assume that such a finding needs further validation and therefore remain cautious in speculating about any potential mechanisms at this stage. Regardless, we discuss this unexpected finding in the Discussion section, calling for future studies for validation.

We have now clarified this in the revised manuscript.

“Results” section (Page 6, Lines 110-111):

“In addition, one negative association was noted between anxiety/stress-related disorders and varicose veins of lower extremities (0.93 [0.89-0.98]).”

“Discussion” section (Page 15, Lines 304-306):

“This also applies to the noted negative association of anxiety/stress-related disorders with varicose veins of lower extremities. With few supportive data from existing literature, such a finding needs to be validated in future studies.”

4. Why only inpatient hospital data in the UKBB data? Outpatient dx codes could also be relevant?

A point of clarity within the genetic determinants method section: the MLM was run against 5 susceptibility scores as a quantitative phenotype? If so, please state clearly.

Authors' response 4:

Thanks for your suggestion. The UK Biobank does not currently contain outpatient care data. In the UK Biobank, the health-related outcomes of the participants were obtained through regular linkage to multiple national databases, including inpatient hospital data, covering all the UK Biobank participants, and primary care data, covering approximately 45% of the participants (Sudlow C et al., PLoS Medicine 2015).

We have now added more details on the available information in the UK Biobank and the present UK cohort in the Methods and Materials and discussed the lack of outpatient care data as one limitation in the Discussion section of the revised manuscript.

“Discussion” section (Page 15, Lines 291-296):

“Several limitations should be acknowledged. First, given the lack of complete primary care data in the Swedish Patient Register and the UK Biobank, as well as the lack of outpatient care data in the UK Biobank, we might have underestimated the number of patients with anxiety and stress-related disorders as well as the number of studied medical conditions, primarily the milder forms of those diseases. Therefore, disease cluster identification based on a more comprehensive data source to validate the findings of the present study is warranted.”

“Methods and Materials” section (Page 19, Lines 374-377):

“The inpatient hospital data, obtained from the Hospital Episode Statistics database, the Scottish Morbidity Record, and the Patient Episode Database, cover all UK Biobank participants since 1997⁶¹. The primary care data, provided by various general practitioner computer system suppliers, cover ~45% of participants since 1985⁶¹.”

For the GWAS analysis, we have now made the suggested changes to clarify that GWAS analyses were conducted to five susceptible scores as quantitative scores separately in the revised manuscript.

“Results” section (Page 8, Lines 139-144):

“To identify the potential genetic determinants for each disease cluster among patients with anxiety or stress-related disorders, we first calculated five cluster-specific quantitative scores as an index of individual’s susceptibility to each disease cluster, and then performed GWAS analyses for the five susceptibility scores separately, among individuals from the UK cohort with eligible genotyping data (n=27,781, Supplementary Fig. 2), using mixed linear model (MLM)-based models.”

“Methods and Materials” section (Page 23, Lines 469-472):

“To assess the association between single nucleotide polymorphisms (SNPs) and five susceptibility scores (as a continuous variable) respectively, we used mixed linear model”

(MLM)-based models for GWAS analyses, adjusted for sex, birth year, genotyping array, and the first 10 principal components (PCs)²⁹.”

5. 22,781 seems a small proportion of the UKBB total. Please account more clearly for the participants excluded.

Authors’ response 5:

Thank you for the comment. As we aimed to identify the genetic determinants for the comorbidity between anxiety/stress-related disorders and the five identified disease clusters, we limited the genetic analyses to the individuals **with anxiety/stress-related disorders** in the UK cohort (N=23,365). In our original manuscript, we provided the flowchart of participant selection in the UK cohort, in the Supplementary Methods and as Supplementary Fig. 2 (see RFigure 1 below). We further excluded 584 individuals after a standard GWAS quality control on the individual-level genotyping data, leaving 22,781 patients with anxiety or stress-related disorders in the genetic analyses.

We have now clarified this in the Methods and Materials section in the revised manuscript.

“Methods and Materials” section (Page 19, Lines 377-384):

“We first excluded individuals who had withdrawn from the UK Biobank (n=108) or with conflicting information (n=1), leaving 502,398 eligible participants (Supplementary Fig. 2). Among these, we constructed a matched cohort, including patients with newly inpatient/primary care diagnosed anxiety or stress-related disorders between January 1, 1997 and December 31, 2019 (N=23,365) who had no history of severe somatic diseases or other psychiatric disorders, and up to ten unaffected individuals for each patient who were randomly selected and individually matched by sex and year of birth using incidence density sampling (N=233,596).”

“Methods and Materials” section (Page 23, Lines 466-469)

“After standard GWAS quality control on the individual level genotyping data²⁸ (Supplementary Methods Section 2.1), we included 22,781 patients (out of the 23,365 patients) and 13,225,429 variants for further analysis.”

We have now added an additional box in the flowchart (RFigure 1 below and updated in Supplementary Fig. 2).

RFigure 1. Flow chart of participant selection in the UK cohort

6. A large number of hits were identified in the GWAS relative to the number of participants (22,781 patients in UKBB if I'm understanding). Current studies of anxiety (Otowa, Purves, Meier, Levey etc) haven't found this many in larger samples. How much of these genetic determinants come from the somatic disease co-morbidities rather than the cluster itself? Meaning, does the cardiovascular cluster 4 discovery mostly show genetic associations with cardiovascular phenotypes? Is the over-representation of loci discovered in cluster 2 and 3 for eye and ear disease surprising in that context?

Authors' response 6:

Thank you for this important comment. Although no comparable genetic results for disease clusters are available in the existing literature, we do find some evidence from previous genetic studies focusing on those comorbidities individually. To test whether these identified cluster-specific genes have been previously associated with any of the individual diseases in each disease cluster, we first conducted hypergeometric tests using the website platform FUMA. For example, we found that a few genes (e.g., *CHDH* and *FILIP1L*) from the cluster featured by eye diseases were associated with ocular axial length, as reported in previous studies. We further searched the top 20 genes of each disease cluster in GeneCards (a gene database providing all annotated and predicted human genes, <https://www.genecards.org/>) and found that several identified cluster-specific genes have been previously associated with relevant psychiatric or somatic traits in the disease cluster.

We have now clarified the identified genes with their associated diseases in RTable 1 below and in Supplementary Table 12 in the revised manuscript.

RTable 1 Mapped genes and their reported cluster-related traits from FUMA and GeneCards

Disease clusters	FUMA-based mapping*		GeneCards-based mapping#	
	Reported traits	Associated genes	Reported traits	Associated genes
Cluster 1 predominated by psychiatric disorders	-	-	Mood disorder	AMOTL1
			Attention deficit hyperactivity disorder (ADHD) and conduct disorder	AMOTL1 , CWC15 , KDM4D
Cluster 2 predominated by eye diseases	Ocular axial length	CHDH , IL17RB , ACTR8 , FILIP1L , LAMA2 , ARHGAP18	Cataract	DMRTA2 , FAF1 , CDKN2C , C1orf185 , STXBP6
			Corneal topography	STXBP6
			Ocular sarcoidosis	STXBP6 , NOVA1
			Diabetic maculopathy	RGS13
			Macular degeneration	KCNT2
Cluster 3 predominated by ear diseases	-	-	-	-
Cluster 4 predominated by cardiovascular diseases	-	-	Atrial fibrillation	EFNA5 , PRPF38B
			Coronary artery disease	PRPF38B
			Stroke	PRPF38B
			Myocardial infarction	PRPF38B
			Angina pectoris	PRPF38B
			Heart failure	PRPF38B
			Cardioembolic stroke	COL23A1
			Diastolic blood pressure	CCNH
Cluster 5 predominated by skin and genitourinary diseases	-	-	Moderate albuminuria	ABHD4 , OXA1L , LRP10 , REM2
			Glomerular filtration rate	RBM23 , PRMT5 , CDH24

* Conducted by hypergeometric tests using the website platform FUMA

Searched in the 'function' section of GeneCards using the top 20 identified genes for each disease cluster.

“Results” section (Page 9, Lines 156-159):

“Based on information from FUMA and GeneCards, we found that several cluster-specific genes have been previously associated with individual psychiatric or somatic traits in the disease cluster (e.g., *PRPF38B* for angina pectoris and myocardial infarction, Supplementary Table 12).”

“Discussion” section (Page 13, Lines 248-257):

“For instance, some mapped genes for susceptibility to the disease cluster featured by psychiatric disorders (e.g., AMOTL1, CWC15, KDM4D) were reported to be associated with mood disorders⁴⁴ as well as attention deficit hyperactivity disorder (ADHD) and conduct disorder⁴⁵. IQCB1 and GOLGB1, the genes identified for the disease cluster dominated by eye diseases, have been associated with corneal resistance factor (a measure of the biomechanical properties of the cornea)⁴⁶, while CHDH and FILIP1L have been associated with ocular axial length⁴⁷. Additionally, some mapped genes for the disease cluster dominated by cardiovascular disease have been demonstrated as risk genes for diastolic blood pressure (COL23A1 and PHYKPL)⁴⁸ or stroke (COL23A1, PHYKPL, and ADAMTS2)^{49,50}.”

“Methods and Materials” section (Page 24, Lines 481-487):

“To test whether these identified cluster-specific genes are associated with any of the individual diseases in each disease cluster, we first conducted hypergeometric tests using the function of ‘GENE2FUNC’ in the website platform ‘FUMA’. As an alternative approach serving a similar purpose, we searched GeneCards, a gene database providing all annotated and predicted human genes (<https://www.genecards.org/>), to identify traits that have been previously associated with the top 20 identified genes of each disease cluster.”

Regarding the comment that more genetic hits were identified in our study in comparison to other studies focusing on genetic determinants of anxiety, **we find it challenging to compare the results as they studied two different phenotypes** (i.e., we focus on disease clusters after anxiety/stress-related disorder, not anxiety/stress-related disorder *per se*).

It is absolutely possible that these new genes may come from comorbidities, independent of prior anxiety/stress-related disorders. To explore this possibility, we additionally conducted GWAS analysis for each disease cluster among individuals without anxiety/stress-related disorders (n=452,148) and then compared its mapped genes to the results of the corresponding GWAS among individuals with anxiety/stress-related disorders (n=27,781). The results are now shown in RTable 2 below and Supplementary Table 15 in the revised manuscript. In brief, we found only a small proportion of overlapping genes (i.e., 5 and 2 genes for clusters predominated by ear diseases and cardiovascular disease, respectively) in both groups, indicating that only a few identified genes were driven by the disease cluster (i.e., comorbidities). In addition, we postulate that the higher number of hits observed among individuals with anxiety/stress-related disorders, compared to GWAS results of individuals without anxiety/stress-related disorders, could be due to a more specific phenotype, with larger biological similarities, was identified in the former group, e.g., multiple mechanisms have been involved in the development of CVDs (further differing by subtypes of CVDs), but in the current study, we merely identified CVD cases that originate from biological dysfunctions induced by prior anxiety/stress-related disorders and thereby with clearer and more uniform biological underpinnings. This may particularly apply to clusters 2 and 3, where we found more genetic hits for the comorbid status (i.e., among individuals with anxiety/stress-related disorders) than for the specific disease cluster only (i.e., among individuals without anxiety/stress-related disorders).

We have now clarified this in the Results and Methods and Materials sections in the revised manuscript.

“Results” section (Page 9, Lines 175-180):

“Furthermore, comparing the mapped genes for each disease cluster among individuals with anxiety/stress-related disorders (n=27,781) to those obtained among individuals without anxiety/stress-related disorders (n=452,148, Supplementary Table 15), we found only a small proportion of overlapping genes (i.e., 5 and 2 genes for clusters predominated by ear diseases and cardiovascular disease, respectively), indicating that few identified genetic hits were driven by the disease clusters only.”

“Methods and Materials” section (Page 24, Lines 490-494):

“Additionally, to test whether the identified genes were primarily driven by the studied disease clusters, independent of the prior anxiety/stress-related disorders, we conducted additional GWAS analyses for those disease clusters among individuals without anxiety/stress-related disorders (n=452,148).”

RTable 2 Number of mapped genes in GWAS of disease clusters among individuals with/without anxiety/stress-related disorders

Disease clusters	Number of mapped genes among individuals with anxiety/stress-related disorders (n=22,781)	Number of mapped genes among individuals without anxiety/stress-related disorders (n=452,148)	Number of common genetic genes	Top 5 reported traits based on GeneCards*
Cluster 1 predominated by psychiatric disorders	20	173	0	
Cluster 2 predominated by eye diseases	362	244	0	
Cluster 3 predominated by ear diseases	440	13	5	C8orf76 Body height ZHX1 - C8ORF76 ZHX1 3-(3-amino-3-carboxypropyl) uridine measurement TATDN1 Triglyceride measurement, alcohol drinking, alcohol consumption measurement, total cholesterol measurement, red blood cell distribution width NDUFB9 Level of deoxyribonuclease tatdn1 in blood serum, educational attainment, peripheral arterial disease, traffic air pollution measurement, multiple sclerosis
Cluster 4 predominated by cardiovascular diseases	30	2165	2	DRAM2 Adolescent idiopathic scoliosis, mean platelet volume, fev/fec ratio, chronic obstructive pulmonary disease, platelet count CEPT1 Adolescent idiopathic scoliosis, mean platelet volume, fev/fec ratio, chronic obstructive pulmonary disease, serum gamma-glutamyl transferase measurement
Cluster 5 predominated by skin and genitourinary diseases	229	12	0	

* Searched in 'function' section of GeneCards

7. I don't see any report of how inflation was accounted for in the GWAS. Please report lamda, intercept and attenuation ratio of ld score regression in main text.

Authors' response 7:

Thank you for the comment. We have now added the suggested LD score regression analysis and reported the LDSC results in RTable 3 below and in Supplementary Table 11 in the revised manuscript.

RTable 3 LD score regression of GWAS for each disease cluster subsequent to anxiety and stress-related disorders

Disease clusters	λ_{GC}	Intercept (SE)	Attenuation ratio
Cluster 1 predominated by psychiatric disorders	1.0225	0.9914 (0.0062)	< 0
Cluster 2 predominated by eye diseases	1.0105	0.9987 (0.0058)	< 0
Cluster 3 predominated by ear diseases	1.0007	1.0023 (0.0073)	3.1828
Cluster 4 predominated by cardiovascular diseases	1.0225	0.9827 (0.0075)	< 0
Cluster 5 predominated by skin and genitourinary diseases	1.0135	0.9951 (0.0058)	< 0

λ_{GC} : genomic control inflation factor; SE: standard error

We have now clarified this in the Results and Methods and Materials sections in the revised manuscript.

“Results” section (Page 8, Lines 148-150):

“According to the genomic inflation analysis results of Linkage Disequilibrium (LD) score regression, we found little indication for confounding effects in the GWAS of the five disease clusters (Supplementary Table 11).”

“Methods and Materials” section (Page 24, Lines 473-473):

“The inflation of GWAS analyses was tested by LD score regression⁶².”

8. This is a novel and interesting study. I'd like a little more detail on methodology and more clarity on what the results of the genetic determinant analysis means (are these truly co-morbidity markers or would we see these in a univariate analysis of the co-morbid disease trait alone)?

Authors' response 8:

Thank you for this comment. We have now added more statements to clarify that these genetic determinants can be considered as markers for the co-occurrence of anxiety/stress-related disorders and the five identified disease clusters (see below for the amendments we have made in the revised manuscript). Please also see the Authors' response 6 above, which might help

explain the difference between our phenotypes and the comorbid traits that were defined in other studies.

“Results” section (Page 8, Lines 139-144):

“To identify the potential genetic determinants for each disease cluster among patients with anxiety or stress-related disorders, we first calculated five cluster-specific quantitative scores as an index of individual’s susceptibility to each disease cluster, and then performed GWAS analyses for the five susceptibility scores separately, among individuals from the UK cohort with eligible genotyping data (n=27,781, Supplementary Fig. 2), using mixed linear model (MLM)-based models.”

Reviewers #2:

Comments:

9. The present study aimed to determine disease clusters after a diagnosis of anxiety or stress-related disorders and the to identify the genetic basis of these. The study is very comprehensive, both conceptually and methodologically, and it presents with a novel approach towards the study of comorbidity. Nevertheless, several aspects need to be further clarified so as to make the added value of the study more visible.

Authors’ response 9:

Thanks for your positive comment. We have made further clarifications in the revised manuscript, as suggested.

10. The authors state that the cluster approach to comorbidity analysis provides a more comprehensive understanding of phenotype heterogeneity in persons with anxiety or stress-related disorders. However, the five clusters identified are truly system/organ-specific and reveal little novelty with respect to the single-disease associations mentioned in previous literature. Besides the limited novelty, these findings contrast with previous literature on patterns of multimorbidity, whereby anxiety was often part of complex multi-system/organ clusters. Related to this, the authors provide little or no comparative insights with previous studies on patterns of multimorbidity in comparable age groups.

Authors’ response 10:

Thank you. We agree that many medical conditions from the same disease cluster belong to the same organ/system (e.g., cluster 2 and cluster 3), possibly due to biological relatedness. However, we managed to cluster diseases across biological systems as well, considering the temporal order and high intrinsic connectivity between diseases, and with validation across two populations. Specifically, we identified cluster 4, including several cardiovascular diseases and respiratory diseases, and cluster 5, including skin diseases and genitourinary diseases. In addition, to the best of our knowledge, we showed for the first time the connectivity of certain diseases from the same organ/system in relation to anxiety/stress-related disorders, which has not been noted in previous research. Therefore, we believe that our study provides a

comprehensive picture of comorbidities and their genetic basis in relation to anxiety and stress-related disorders. We now added more statements to highlight the novelty of our study in terms of methodology and findings.

“Discussion” section (Page 11, Lines 213-216):

“As a result, we managed to identify five key clusters of disease associated with a prior diagnosis of anxiety or stress-related disorders (with component diseases in the same system or across different systems), considering the temporal order and high intrinsic connectivity between diseases and with validation across two populations.”

“Discussion” section (Page 15, Lines 289-290):

“This analytic strategy could be applied as a pipeline for studying comorbidities of other phenotypes.”

Moreover, although we completely agree with the reviewer that we should compare the comorbidity patterns to previous studies, there are no directly comparable data. A recent study showed an overall disease trajectory picture in the entire Danish population and established a browser (Siggaard T et al., Nature Communications, 2020). By searching for disease trajectories in relation to anxiety and stress-related disorders (ICD-10: F40, F41, F43) using this browser, we found that some key diseases in our study, such as cardiovascular diseases that were identified *subsequent to* stress-related disorders, were listed *prior to* stress-related disorders in the browser. **However, the inconsistent results observed between our and Siggaard T’s studies reflect different research questions with different methodological considerations.** We aimed to disentangle comorbidity clusters *after* anxiety/stress-related disorders, while Siggaard T et al. examined diseases occurring both *before* and *after* anxiety/stress-related disorders (and any other disease). To rule out the impact of preexisting diseases on the subsequent disease clusters, we excluded anxiety/stress-related disorder patients with preexisting severe somatic diseases at the time of diagnosis in our analyses, which is not the case in Siggaard T et al.’s work. We have now clarified this in the revised manuscript.

“Discussion” section (Page 12, Lines 236-242):

“Prior attempts to illustrate disease networks include a recent Danish study based on national data from inpatient and outpatient care, which established a browser presenting the disease trajectories both before and after a target disease of interest⁴³. Using this browser, we found that some key diseases in our study, such as cardiovascular diseases identified subsequent to stress-related disorders, were listed prior to stress-related disorders. However, as the comparability of these two studies is limited (due to the different research purposes and study designs), the inconsistent results do not necessarily invalidate each other.”

11. Even if the authors mention the limitation linked to the lack of primary care data coverage, they should further elaborate on the sensitivity and specificity of Swedish NPR and UK Biobank data to capture anxiety or stress-related disorders. I would assume that the biggest part of these diagnoses are made in primary care, and may never reach secondary/tertiary care. This has important implications in terms of sample selection bias that requires further careful and

critical elaboration.

Authors' response 11:

Thank you for this comment. The absence of data from primary care is indeed one of our limitations acknowledged in this study. As noted, we have clarified throughout the text that we studied anxiety/stress-related disorders diagnosed through specialized care, resulting in an underestimated number of patients with relatively mild-to-moderate forms of anxiety/stress-related disorders, as well as the studied medical conditions. We have therefore further highlighted that our findings should be interpreted with caution when generalized to other groups of patients or other settings.

Apart from the lack of primary care data, the specialized diagnosis of anxiety/stress-related disorders is considered to be reliable. The Swedish National Patient Register (NPR) covers all inpatient care since 1987 and >80% outpatient care since 2001, with the diagnostic codes validated for most common diseases through external review (positive predicted value [PPV] between 85-95% (Ludvigsson JF et al., BMC Public Health 2011). Specifically, the diagnosis of social anxiety disorder was validated in the NPR and considered to be reliable (PPV: 81%) (Vilaplana-Pérez A et al., BMC Psychiatry 2020). The diagnosis of stress-related disorders was also considered to be reliable based on a validation study assessing the validity of post-traumatic stress disorder (PTSD) (Hollander AC et al., BMJ Open 2019). Although there is currently no validation of diagnoses of anxiety or stress-related disorders in the Hospital Episode Statistics in the UK Biobank, the accuracy of depression diagnoses is good (PPV: 73%, Davis KA et al., PLoS One 2018).

We now have clarified this limitation and emphasized that we focused on the anxiety/stress-related disorders attended in specialized care throughout the revised manuscript.

“Abstract” section (Page 3, Lines 43-46):

“These findings motivate further mechanistic explorations and aid early risk assessment for cluster-based disease prevention among patients with newly diagnosed anxiety/stress-related disorders in specialized care.”

“Discussion” section (Page 15, Lines 291-296):

“Several limitations should be acknowledged. First, given the lack of complete primary care data in the Swedish Patient Register and the UK Biobank, as well as the lack of outpatient care data in the UK Biobank, we might have underestimated the number of patients with anxiety and stress-related disorders as well as the number of studied medical conditions, primarily the milder forms of those diseases. Therefore, disease cluster identification based on a more comprehensive data source to validate the findings of the present study is warranted.”

“Discussion” section (Page 16, Lines 309-313):

“In conclusion, based on detailed phenotypic and genetic analyses of two large-scale cohorts, we identified five distinct disease clusters subsequent to an inpatient/outpatient diagnosis of anxiety and stress-related disorders, featured by other psychiatric disorders, eye diseases, ear

diseases, cardiovascular diseases, and skin and genitourinary diseases as predominant diseases in each cluster.”

“Methods and Materials” section (Page 18, Lines 355-363):

“In the Swedish cohort, we defined anxiety or stress-related disorders as any first specialist care diagnosis in an inpatient or outpatient hospital visit, where these disorders were identified as the primary discharge diagnosis, according to the Swedish Patient Register, using the 10th Swedish revision of the International Classification of Diseases (ICD-10) codes (anxiety: F40 and F41, stress-related disorder: F43) (Supplementary Table 1).”

“Methods and Materials” section (Page 18, Lines 363-366):

“The diagnostic codes for most common diseases in the Patient Register have been validated, showing a satisfactory accuracy with positive predicted values [PPV] of 85-95% for most common diseases²⁴, 81% for social anxiety disorder⁵⁸, and 75-90% for PTSD⁵⁹.”

12. It is not entirely clear to me why and how the disease trajectory (temporal relationship) and comorbidity network (non-temporal relationship) analyses are combined. It would be helpful if the authors could provide some real examples in the methods as to how this was performed. Related to this, did the authors not think of other, more straightforward and person-centered clustering approaches such as latent class analysis?

Authors’ response 12:

Thank you for this comment. Disease trajectory and comorbidity network analysis are both validated methods that have been designated to explore complex disease networks (Hidalgo CA et al., PLoS Computational Biology 2020; Siggaard T et al., Nature Communications 2020) and have been widely used in previous studies, including papers from our group (Han X et al., Molecular Psychiatry 2021; Shen Q et al., eClinicalMedicine 2023; Hou C et al., American Journal of Clinical Nutrition 2021). These two data-driven methods have both advantages and disadvantages. For instance, disease trajectory analysis can visualize directional disease networks according to the temporal order of cooccurring diseases, which, however, requires reliable data on the time of disease onset. In contrast, comorbidity network analysis can identify disease clusters with high intrinsic connectivity of diseases. However, without the consideration of temporality, the identified disease clusters are less informative and less stable, presenting as changes in component diseases when using different disease selection strategies or different study samples.

In our study, as we ultimately wanted to explore the genetic basis of disease clusters associated with anxiety/stress-related disorders using GWAS, the most important purpose of the phenotypic analyses was to provide reliable definitions of the studied ‘disease clusters’. Therefore, we combined those two methods, which enabled a theoretically cross-validation of identified disease clusters and their key component diseases using both theories of temporality and intrinsic connectivity. To make it clearer, we made a figure to illustrate the process of how the results from those two analyses were combined (see RFigure 1 below and Supplementary Fig. 10 in the revised manuscript).

As stated above, our aim is to identify robust disease clusters with clear temporal order and high intrinsic connectivity subsequent to anxiety/stress-related disorders, and latent class analysis is not able to achieve this research purpose.

We have clarified the rationale and process of combining those two analyses and provided an example in the revised manuscript:

“Methods and Materials” section (Page 22, Lines 438-451):

“As disease trajectory analysis is designed to visualize sequential disease progression while comorbidity network analysis captures disease groups with high intrinsic connectivity, the combined use of those two data-driven approaches can theoretically lead to the identification of more reliable disease clusters (i.e., groups of diseases with both temporal and non-temporal relationships). Thus, based on results from the aforementioned disease trajectory and comorbidity network analyses, we defined disease clusters as the first layer diseases (D1) and their subsequent diseases in a disease trajectory that were also located within the same comorbidity module (Supplementary Fig. 10). For example, in the disease trajectories, out of all diseases in the first layer, “F32” and “F10” were located in one comorbidity module (i.e., the module predominated by psychiatric disorders) derived from the comorbidity network. We then found the following diseases of “F32” and “F10” in the trajectories which were also in such a comorbidity module to constitute a disease cluster (i.e., the disease cluster featured by psychiatric disorders including “E66”, “F10”, “F13”, “F19”, “F20”, “F30”, “F32”, “F39”, “F60”, and “F90”).”

RFigure 1 Identification of disease cluster featured by psychiatric disorders by combination of disease trajectory and comorbidity network

13. Bearing in mind that anxiety or stress-related disorders are, to a great extent, socially patterned, stratification by socio-economic status would be of high interest and added value. The authors exclude non-Swedish born subjects from their Swedish sample; this decision needs to be better justified, precisely because the immigrant layer is one that strongly intersects with social class.

Authors' response 13:

Thank you! We agree that socio-economic status is an important aspect to take into account when studying comorbidities in relation to anxiety and stress-related disorders. However, in the Swedish cohort, we only limited information about socio-economic status (i.e., the highest level of education and income at the year of index date obtained from the Swedish Longitudinal Integration Database for Health Insurance and Labor Market). During the PheWAS analyses, we controlled for the effect of socio-economic status on the studied association by adjusting for highest education and income in the Cox models (see Supplementary Methods). Furthermore, as disease trajectory and comorbidity network analyses require a large sample size to ensure enough data power, performing further stratification analysis by socio-economic status is not feasible in the present study. The possible modification effect of socio-economic status on the disease clusters associated with anxiety/stress-related disorders, as well as their genetic determinants, needs to be addressed in future studies. We have clarified this in the Methods section in the revised manuscript.

In addition, because we aimed to further explore the genetic determinants underlying the identified comorbidity clusters, our analysis was therefore restricted to participants with similar genetic backgrounds. The exclusion of non-Swedish-born individuals was also to minimize the potentially different health-seeking behaviors between Swedish-born and non-Swedish-born individuals. We have now added the motivation for excluding non-Swedish-born individuals in the revised manuscript.

“Methods and Materials” section (Page 21, Lines 411-414):

“We performed a PheWAS to investigate the associations between anxiety or stress-related disorders and each medical condition (Supplementary Method Section 1.1), using Cox regression models stratified by matching variables (i.e., sex and birth year) with adjustment for highest education and income.”

“Methods and Materials” section (Page 17, Lines 334-341):

“By cross linkage to the Total Population Register using the unique Swedish personal identification numbers, we included all Swedish-born individuals residing from 2001 to 2016 in Sweden and excluded those with any pre-existing psychiatric disorders or history of severe somatic diseases determined by the Charlson Comorbidity Index (CCI) before 200125, leading to a study population of 8,456,485 (Supplementary Fig. 1). We focused on Swedish-born individuals in the present study to reduce the heterogeneity in genetic background as well as other sociodemographic factors, including differential health-seeking behaviors.”

14. On a more general level, I would like to inquire the authors concerning the following

justification for running one of the objectives in their study. They mention that “the identification of cluster-specific, instead of diseases-specific, genetic variants, have potential to increase the feasibility of developing early intervention for preventing a general health decline among patients with those psychiatric disorders”. The authors should further and critically elaborate on their views of how gene-based risk estimates could help in the design of targeted preventive interventions.

Authors’ response 14:

Thank you for the comment. As the reviewer has correctly pointed out, the aim of the present study is, using gene-based risk estimates, to identify groups of patients with anxiety/stress-related disorders who are further at risk of developing five distinct disease clusters. We assume our findings could aid the development of preventive interventions in the following aspects. First, as primary prevention is to prevent diseases that have not occurred (but possibly happen in the future), interventions serving such a purpose need to be harmless and of high necessity. For the case of anxiety and stress-related disorders that are linked with not only one but also a sequence of multiple subsequent diseases, disease-based preventive strategies are obviously impossible. Thus, in addition to providing biomarkers for risk stratification, the cluster-based genetic determinants identified in our study are of great importance in terms of pinpointing the potential biological pathways that could ultimately be used to effectively prevent the development of multiple diseases with biological similarities. Second, those cluster-based genetic determinants, if verified in experimental studies, can be used for matching existing medications that have been proven safe and effective for primary prevention purposes.

We have now clarified this justification in the revised manuscript.

“Introduction” section (Page 5, Lines 75-79):

“Furthermore, with the notion that disease located in the same cluster should have shared or linked biological mechanisms, the identification of cluster-specific, instead of diseases-specific, genetic variants, has the potential to realize the prevention of a general further health decline among patients with anxiety or stress-related disorders.”

“Discussion” section (Page 14, Lines 276-279):

“For instance, medications indicated for the genes/pathways within each disease cluster could be further tested for effectiveness in reducing risks of further disease development among individuals with anxiety and stress-related disorders.”

REVIEWERS' COMMENTS

Reviewer #1 (Remarks to the Author):

I am satisfied with the response to my queries. They were thorough and I think it's reasonable, given the challenge to access the data.

I think they were overly conciliatory in the statement "Last, our findings cannot be generalized to other populations of non-European ancestry or different health care systems than in Sweden and the UK"

"Cannot be generalized" is too strong a statement, more that they may not be generalized.

Reviewer #2 (Remarks to the Author):

The authors have adequately addressed all of my concerns. I am thankful and appreciate their hard and rigorous work.

Responses to the comments of the referees*:

*All page and line numbers correspond to the page and line numbers of the **‘Revised Manuscript_clean’** version.

Comments from Reviewers

Response to reviewers #1:

I am satisfied with the response to my queries. They were thorough and I think it's reasonable, given the challenge to access the data.

Authors' responses: Thank you for the positive comments!

I think they were overly conciliatory in the statement "Last, our findings cannot be generalized to other populations of non-European ancestry or different health care systems than in Sweden and the UK". "Cannot be generalized" is too strong a statement, more that they may not be generalized.

Authors' responses: Thanks. We have made the suggested changes in the revised manuscript.

‘Discussion’ section (Page 15, Lines 304-306):

‘Last, our findings may not be generalized to other populations with non-European ancestry or different health care systems than in Sweden and the UK.’

Response to reviewers #2:

The authors have adequately addressed all of my concerns. I am thankful and appreciate their hard and rigorous work.

Authors' responses: Thank you!